# The CCCH-Type Zinc-Finger Protein GhC3H20 Enhances Salt Stress Tolerance in *Arabidopsis thaliana* and Cotton through ABA Signal Transduction Pathway

**DOI:** 10.3390/ijms24055057

**Published:** 2023-03-06

**Authors:** Qi Zhang, Jingjing Zhang, Fei Wei, Xiaokang Fu, Hengling Wei, Jianhua Lu, Liang Ma, Hantao Wang

**Affiliations:** State Key Laboratory of Cotton Biology, Institute of Cotton Research, Chinese Academy of Agricultural Sciences, Anyang 455000, China

**Keywords:** cotton, *GhC3H20*, salt stress, *GhPP2CA* and *GhHAB1*, ABA signaling pathway

## Abstract

The CCCH zinc-finger protein contains a typical C3H-type motif widely existing in plants, and it plays an important role in plant growth, development, and stress responses. In this study, a CCCH zinc-finger gene, *GhC3H20*, was isolated and thoroughly characterized to regulate salt stress in cotton and *Arabidopsis*. The expression of *GhC3H20* was up-regulated under salt, drought, and ABA treatments. GUS activity was detected in the root, stem, leaves, and flowers of Pro*GhC3H20*::GUS transgenic *Arabidopsis*. Compared with the control, the GUS activity of Pro*GhC3H20*::GUS transgenic *Arabidopsis* seedlings under NaCl treatment was stronger. Through the genetic transformation of *Arabidopsis*, three transgenic lines of 35S-*GhC3H20* were obtained. Under NaCl and mannitol treatments, the roots of the transgenic lines were significantly longer than those of the wild-type (WT) *Arabidopsis*. The leaves of the WT turned yellow and wilted under high-concentration salt treatment at the seedling stage, while the leaves of the transgenic *Arabidopsis* lines did not. Further investigation showed that compared with the WT, the content of catalase (CAT) in the leaves of the transgenic lines was significantly higher. Therefore, compared with the WT, overexpression of *GhC3H20* enhanced the salt stress tolerance of transgenic *Arabidopsis*. A virus-induced gene silencing (VIGS) experiment showed that compared with the control, the leaves of pYL156-*GhC3H20* plants were wilted and dehydrated. The content of chlorophyll in pYL156-*GhC3H20* leaves was significantly lower than those of the control. Therefore, silencing of *GhC3H20* reduced salt stress tolerance in cotton. Two interacting proteins (GhPP2CA and GhHAB1) of GhC3H20 have been identified through a yeast two-hybrid assay. The expression levels of *PP2CA* and *HAB1* in transgenic *Arabidopsis* were higher than those in the WT, and pYL156-*GhC3H20* had expression levels lower than those in the control. *GhPP2CA* and *GhHAB1* are the key genes involved in the ABA signaling pathway. Taken together, our findings demonstrate that GhC3H20 may interact with GhPP2CA and GhHAB1 to participate in the ABA signaling pathway to enhance salt stress tolerance in cotton.

## 1. Introduction

Cotton is an important oil crop and fiber crop in China, which plays an important role in the national economy [1]. Soil salinization affects not only cotton yield but also cotton quality. It is essential to study the mechanism of upland cotton in response to salt stress, which is closely linked to cotton production and the agricultural economy in China [2]. Transcription factors can activate the expression of downstream genes and form a transcriptional regulatory network in response to stress [3,4]. Transcription factors such as the AP2/ERF family members related to plant stress can regulate the expression of multiple functional genes related to plant stress. It is more effective to use transcription factors to study the stress resistance of crops and cultivate good stress-resistant varieties than to use single genes for the genetic improvement of crops [5].

The zinc-finger transcription factor family, as one of the largest transcription factor families in plants, plays an important role in multiple biological processes, such as morphogenesis, signal transduction, and environmental stress responses [6,7]. Zinc-finger transcription factors contain zinc-finger motifs in which cysteines and/or histidines coordinate with a few zinc atoms to form the local peptide structures that are essential for their specific functions [8]. Several plant zinc-finger families, such as the RING-finger, ERF, WRKY, DOF, and LIM families, regulate gene expression with the aid of DNA-binding, protein-binding proteins, or RNA-binding proteins [9,10,11,12,13]. According to their structural diversities, the zinc-finger transcription factor family has been classified into nine types: C2H2, C8, C6, C3HC4, C2HC, C2HC5, C4, C4HC3, and CCCH [6,8,14,15,16]. The CCCH family contains a typical C3H-type motif, and members of this family have already been identified in organisms from yeast to humans [8,9,10]. In plants, multiple CCCH zinc-finger proteins were found to be involved in abiotic and biotic stresses. For example, as a nuclear CCCH zinc-finger protein, overexpressed *GhZFP1* enhanced salt stress and fungal disease tolerance in transgenic tobacco plants by interacting with GZIRD21A and GZIPR5 [17]. The CCCH zinc-finger members participated in salt stress via the ABA signaling pathway. Overexpressed *AtOZF2 Arabidopsis* plants were insensitive to salt stress, and the mutant *atozf2 Arabidopsis* plant significantly reduced salt stress tolerance. Further study showed that *AtOZF2* regulated salt stress via the ABA signaling pathway mediated by *ABI2* [18]. Under salt stress, the expression of three salt stress-responsive genes (*RAB18*, *COR15A*, and *RD22*) in the ABA-dependent pathway was significantly higher in *AtC3H17* OXs than in the WT. The results indicated that the CCCH zinc-finger gene *AtC3H17* may be involved in salt stress by the ABA-dependent pathway [19]. As a nuclear transcriptional activator, IbC3H18 interacted with IbPR5 and enhanced salt and drought stress tolerance in transgenic tobacco plants by regulating the expression of a range of abiotic stress-responsive genes involved in reactive oxygen species (ROS) scavenging, ABA signaling, photosynthesis, and ion transport pathways [20]. Pieces of evidence show that CCCH zinc-finger genes are involved in response to salt stress through the ABA-dependent pathway.

In this study, based on the transcriptome data related to salt stress [21], an up-regulated CCCH zinc-finger gene *GhC3H20* was identified. qRT-PCR results showed that *GhC3H20* was up-regulated under NaCl, PEG, and ABA treatments. Compared with the control, the GUS activity of Pro*GhC3H20*::GUS transgenic *Arabidopsis* seedlings under NaCl treatment was stronger. Overexpression of *GhC3H20* in *Arabidopsis* could promote salt stress tolerance, and silencing of *GhC3H20* in cotton could decrease salt stress tolerance. In addition, GhC3H20 could interact with GhPP2CA and GhHAB1 to regulate salt stress by the ABA signaling pathway. This study lays the foundation for future studies of *GhC3H20* in the improvement of salt stress tolerance in cotton.

## 2. Results

### 2.1. Gene Structure, Phylogenetic Analysis, and Protein Sequence Alignment Analysis of GhC3H20

The coding sequences of the *GhC3H20* (*GH_D08G2771*) gene were cloned from the upland cotton material TM-1. The open reading frame of *GhC3H20* was 1062 bp and encoded 353 amino acid residues. The estimated molecular mass (Mw) of the GhC3H20 protein was 39.3 kDa, and the isoelectric point (pI) was 6.23. Subcellular prediction analysis results showed that GhC3H20 might be located in the nucleus of the cell. The result of gene structure analysis showed that *GhC3H20* contained an exon (Figure 1A). The cotton *GhC3H20* gene is homologous to the *Arabidopsis AtC3H20* (*AT2G19810*) gene, and the protein sequence similarity is 56%. In a previous study, phylogenetic analysis result revealed that *AtC3H20* belonged to the CCCH zinc-finger family group IX [22]. The neighbor-joining tree result of *GhC3H20* and CCCH family group IX members in *Arabidopsis* showed *GhC3H20*, *AT4G29190*, *AT2G19810*, and *AT2G25900* divided into one branch (Figure 1B). The protein sequence alignments of GhC3H20, AT4G29190, AT2G19810, and AT2G25900 showed that all four proteins contained two CCCH motifs and most amino acid residues were conserved (Figure 1C).

### 2.2. Expression Pattern Analysis of the GhC3H20 Gene under NaCl and PEG Treatments and in Eight Cotton Tissues

The transcriptome data of salt treatment showed that the *GhC3H20* gene responded to salt stress [21]. Therefore, the *GhC3H20* gene was selected to do qRT-PCR under salt and PEG treatments. qRT-PCR results revealed that the expression levels of *GhC3H20* were the highest at 48 h and 24 h of NaCl and PEG treatments (Figure 2A,B). The results showed that the *GhC3H20* gene was up-regulated under salt and drought stresses. The expression levels of the *GhC3H20* gene in eight tissue samples (vegetative organs: roots, stems, leaves, and buds; reproductive organs: petals, stamens, pistils, and fibers) of upland cotton were determined by qRT-PCR. As shown in Figure 2C, in reproductive organs, the expression levels of the *GhC3H20* gene were highest in stamens, and, among vegetative organs, the expression levels of the *GhC3H20* gene were highest in stems.

### 2.3. Promoter Analysis of GhC3H20

The PlantCare website was employed to analyze cis-acting elements in the 2000 bp promoter of *GhC3H20*. The result revealed that this region included stress-response, light-response, and hormone-response elements. Among hormone-response elements, ABA-response elements occur in the largest number as six elements (Appendix A). ABA (100 μM) was sprayed on the leaves of upland TM-1 cotton seedlings and the expression levels of *GhC3H20* were measured by qRT-PCR. The results showed that after ABA treatment, the expression levels of *GhC3H20* were increased significantly from 0 h to 6 h and decreased from 6 h to 12 h (Figure 3A). The results indicated that the *GhC3H20* gene had responded to ABA.

To understand the tissue expression specificity of *GhC3H20*, the Pro*GhC3H20*:: GUS vector was constructed and genetically transformed into *Arabidopsis thaliana*. The roots, stems, leaves, flowers, and fruit pods of T_1_ generation transgenic *Arabidopsis* plants were taken for GUS staining, respectively. The results showed GUS staining was found in all tissues of transgenic *Arabidopsis* plants, except fruit pods (Figure 3B).

To further understand the promoter of *GhC3H20* in response to salt stress, the GUS staining of transgenic seedlings was detected under NaCl (150 mM) treatment. GUS activity was more strongly expressed in the leaves and stems of NaCl-treated transgenic *Arabidopsis* seedlings than in the control. (Figure 3C,D).

### 2.4. Overexpression of GhC3H20 Enhances Salt and osmotic Stress Tolerance in Transgenic Arabidopsis Seedlings

To further understand the relationship of *GhC3H20* in response to salt stress, the 35S-*GhC3H20* vector was constructed and transformed into *Arabidopsis*. The transgenic *Arabidopsis* were detected by 1/2 MS (+kana) and PCR. The T_3_ generation of transgenic *Arabidopsis* was used for further analysis under salt stress. qRT-PCR results indicated that three transgenic *Arabidopsis* lines were significantly overexpressed relative to the WT (Figure 4A). To further understand the function of the *GhC3H20* gene in response to salt and osmotic stresses during seedling stages, the root length of ten *Arabidopsis* seedlings were measured under 0, 150 mM NaCl, and 200 mM mannitol treatments (Figure 4B). Under NaCl and mannitol treatments, the roots of the transgenic lines were significantly longer than those of the WT (Figure 4C). The results indicated that overexpression of *GhC3H20* could enhance the root growth of transgenic *Arabidopsis* seedlings under salt and osmotic stresses.

### 2.5. Overexpression of GhC3H20 Enhanced the Salt Stress Tolerance of Transgenic Arabidopsis at the Seedling Stage

To analyze the function of the *GhC3H20* gene in response to salt stress at the seedling stage, WT and transgenic *Arabidopsis* were planted in nutrient soil and treated with 400 mM NaCl to observe the phenotype. After salt treatment for 5 days, the leaves of the WT wilted and turned yellow. While the transgenic lines still had green rosette leaves (Figure 5A). To better understand the physiological changes in plants, after salt treatment, the leaves of WT and transgenic *Arabidopsis* were taken, and the activities of CAT were measured. The activities of CAT in the transgenic *Arabidopsis* lines were significantly higher than those in the WT (Figure 5B). All results indicated that overexpression of *GhC3H20* could enhance the salt stress tolerance of *Arabidopsis* plants.

### 2.6. Silencing of GhC3H20 in Cotton Decreased Salt Stress Tolerance

To further analyze the putative function of the *GhC3H20* gene, a VIGS strategy was used to knock down the expression levels of the *GhC3H20* gene in cotton. Cotton standard line TM-1 plants were grown until the third true leaf expanded and was treated with 400 mM NaCl for five days. Figure 6A shows that PYL156-*GhPDS* plants (positive control) had an obvious leaf-whitening phenotype. The leaves of the PYL156-*GhC3H20* (silenced plants) plants wilted and turned yellow, while the leaves of the PYL156 (control plants) plants only slightly shrank due to water loss (Figure 6A). The expression levels of *GhC3H20* determined by qRT-PCR indicated that the gene was effectively silenced (Figure 6B). To understand the physiological changes in plants, after salt treatment, the leaves of the control and the silenced plants were taken, and the content of chlorophyll was measured. The content of chlorophyll in the leaves of the control plants was significantly higher than that of the leaves of the silenced plants (Figure 6C). These results indicated that silencing of the *GhC3H20* gene reduced salt stress tolerance in cotton.

### 2.7. GhC3H20 Interacts with GhPP2CA and GhHAB1

A transcriptional activation assay was performed in the yeast cells. The yeast cells containing pGADT7+pGBKT7-*GhC3H20* (experimental group), pGADT7-large T+pGBKT7-lamin C (negative control), and pGADT7-large T+pGBKT7-p53 (positive control) were transformed into the yeast cells and cultured on SD/−Trp/−Leu and SD/−Trp/−Leu/−His/−Ade medium. The positive control and the experimental group grew well on SD/−Trp/−Leu and SD/−Trp/−Leu/−His/−Ade medium, while the negative control could not, demonstrating that GhC3H20 could autonomously activate the reporter genes in the absence of a prey protein (Figure 7A).

Previous studies have reported that CCCH zinc-finger genes participate in salt stress through the ABA signaling pathway [18,19,20]. Therefore, four key genes (*GhPP2CA*, *GhHAB1*, *GhABF3*, and *GhABI1*) of ABA signal transduction were selected to do the Y2H assay. The coding sequences of *GhPP2CA*, *GhHAB1*, *GhABF3*, and *GhABI1* were cloned into the pGADT7 vector. The constructive vectors with pGBKT7-*GhC3H20* were co-transformed into the yeast strain Y2HGold. The results showed that the yeast cells containing the positive control pGBKT7-p53+pGADT7-largeT and the experimental group pGBKT7-*GhC3H20 *+pGADT7-*GhPP2CA* and pGBKT7-*GhC3H20* +pGADT7-*GhHAB1* could grow well on SD/−Trp/−Leu/-His/−Ade+40 mM 3AT medium and turned blue on SD/−Trp/−Leu/−His/−Ade/X-a-Gal/AbA+40 mM 3AT medium. Yeast containing the negative control, the control group pGADT7+pGBKT7-*GhC3H20*, and the experimental group pGBKT7-*GhC3H20*+ pGADT7-*GhABF3* and pGBKT7-*GhC3H20*+ pGADT7-*GhABI1* could not grow on SD-TLHA+40 mM 3AT medium and turned blue on SD-TLHA+X-α-Gal +40 mM 3AT medium. The results showed that GhC3H20 could interact with GhPP2CA and GhHAB1 but did not interact with GhABF3 and GhABI1 (Figure 7B).

### 2.8. GhC3H20 Increased the Expression Levels of ABA Signaling and Osmotic Stress-Related Genes in Arabidopsis and Cotton

qRT-PCR results indicated that the expression of *GhC3H20* was up-regulated under ABA and PEG treatments. Overexpression of *GhC3H20* enhanced the osmotic stress tolerance in *Arabidopsis* seedlings. Therefore, two interaction genes (*PP2CA* and *HAB1*) of *GhC3H20* and two osmotic stress-related genes (*AtNHX1* and *GhNHX2*) were used to do qRT-PCR in *Arabidopsis* and cotton, respectively. Under salt stress, the expression levels of *AtPP2CA*, *AtHAB1*, and *AtNHX1* in transgenic *Arabidopsis* were higher than those in the WT, except *GhHAB1* in Line 3 (Figure 8A,B), and the expression levels of *GhPP2CA*, *GhHAB1*, and *GhNHX2* in the silenced plants were lower than those in the control plants (Figure 8C,D). The expression of *GhPP2CA* and *GhHAB1* were the highest at 6 h of NaCl treatment (Figure 8E). *PP2CA* and *HAB1* are key genes in ABA signaling transduction. NHX protein is a Na^+^/H^+^ antiporter, which regulates osmotic stress by transporting intracellular Na^+^ to be extracellular under high salt concentrations. It is most likely that *GhC3H20* could enhance salt stress tolerance through the ABA signal transduction pathway and osmotic stress pathway.

## 3. Discussion

Having been cloned in various plants such as *Arabidopsis* [23], rice [24], and sweet potato [20], zinc-finger transcription factors play an important role in the regulation of plant abiotic stress responses. However, the regulatory mechanism of the zinc-finger gene in response to salt stress in cotton remains poorly understood. In our study, a salt-induced gene (*GhC3H20*) from the zinc-finger family was identified by transcriptome data of salt stress [21]. The gene encodes 353 amino acids, with two C3H-type motifs and no intron, which indicated that the *C3H20* gene belongs to the CCCH zinc-finger subfamily [22]. Phylogenetic analysis also revealed that the cotton *GhC3H20* gene belongs to a novel CCCH-type zinc-finger protein subfamily. All members of CCCH zinc-finger group IX in *Arabidopsis* have some characteristics in common, consisting of two CCCH-type motifs. In addition, gene structure analysis indicated that they are all intronless genes [22]. The CCCH zinc-finger family has been reported to be involved in salt stress [25]. More recently, the rice *TZF1* gene has been reported to regulate the expression of many abiotic stress tolerance genes to enhance salt stress tolerance in transgenic *Oryza sativa* L. plants [26]. The first CCCH zinc-finger protein *GhZFP1* in cotton has been identified and functionally characterized. Further research showed that GhZFP1 interacted with GZIRD21A and GZIPR5 to regulate salt stress in cotton [17]. Our study lays the foundation for future studies of the CCCH zinc-finger members in the improvement of salt stress tolerance in cotton.

In the present study, the *GhC3H20* gene was induced by salt, drought, and ABA treatments. The homologous genes of *GhC3H20* in *Arabidopsis*, *AtOZF1*, and *AtOZF2* were also induced by salinity and ABA [18]. The cotton *GhZFP1* gene was induced by salt and drought stresses [17]. The result indicated that *GhC3H20* is likely to be involved in salt stress. To understand the relationship between the *GhC3H20* gene and salt stress, we obtained three transgenic *Arabidopsis* lines through genetic transformation. Overexpressed *GhC3H20* enhanced salt stress tolerance in transgenic *Arabidopsis* seedlings. Silencing of *GhC3H20* decreased the salt stress tolerance in cotton. Plants produce some reactive oxygen species (ROS) under salt stress, such as O^2^, H_2_O_2_, O^2−^, and HO, which are highly active molecules and can cause oxidative damage to proteins, DNA, and lipids [27,28]. CAT is the most important antioxidant enzyme to help remove excess ROS in plants, maintain a low level of ROS, and improve the tolerance of plants under stress [29,30]. In this work, we found that under salt treatment, transgenic *Arabidopsis* had higher CAT activity compared with the WT, suggesting that the *GhC3H20* gene can increase the relevant antioxidant enzymes in transgenic *Arabidopsis* in response to salt stress. Chlorophyll is an important material for the photosynthesis of plants. Under stress, the content of chlorophyll decreases and photosynthesis decreases, thus causing damage to plants. Under salt stress, the control cotton plants had a higher content of chlorophyll compared with silent plants. The expression of genes (*AtNHX1* and *GhNHX2*) related to osmotic stress was higher in transgenic *Arabidopsis* than in the WT and was lower in the silenced plants than in the control plants. It is likely that stress-inducible genes were induced by the expression of *GhC3H20*. The increased abundance of *GhC3H20* transcripts probably up-regulated the transcription of several stress-inducible genes, which in turn contributed to increased endurance under the stress conditions. It has been reported that overexpression of the CCCH-tandem zinc-finger protein OsTZF1 promotes salt stress tolerance by inducing the expression of some stress-related genes in Ubi:*OsTZF1* OX plants [31]. These results demonstrated that *GhC3H20* might participate as a positive transcript factor in the regulation of salt stress.

As an important signaling molecule, ABA plays a key role in abiotic stresses such as salt stress [32,33,34]. Under stress, ABA can improve plant tolerance by altering stomatal closure, modulating root architecture, and osmolyte biosynthesis [35,36]. Some genes enhance plant tolerance through the ABA signaling pathway or the ABA biosynthesis pathway [37,38]. What is more, the expression levels of the stress-related genes were decreased in mutants associated with defects in ABA biosynthesis or response. For example, under salt stress, the expression levels of the *RD29B* gene in *ABA1* and *ABI1* mutants are extremely low or do not expressed [39]. All evidence suggests that ABA is a key hormone in response to stress. Multiple studies have reported that CCCH zinc-finger proteins participate in salt stress via the ABA signaling pathway [18,19,20]. In our study, the cotton *GhC3H20* gene was up-regulated under ABA treatment. To elucidate the precise molecular mechanism of GhC3H20 increasing salt stress tolerance in transgenic plants and cotton, two proteins–GhPP2CA and GhHAB1–that interact with GhC3H20 were isolated and identified by Y2H assay. The expression of *GhPP2CA* and *GhHAB1* was induced by salt stress. Under salt stress, the expression levels of *AtPP2CA* and *AtHAB1* in transgenic *Arabidopsis* plants were higher than those in WT *Arabidopsis*, except *AtHAB1* in Line 3, and the expression levels of *GhPP2CA* and *GhHAB1* in silenced plants were lower than that in control plants. *GhPP2CA* and *GhHAB1* genes were likely induced by the expression of *GhC3H20*. The increased abundance of *GhC3H20* transcripts probably up-regulated the transcription of *GhPP2CA* and *GhHAB1* genes, which in turn contributed to increased endurance under salt stress. *AtOZF2* was reported to enhance salt stress tolerance by increasing and decreasing the expression of the *AtABI2* gene in transgenic *Arabidopsis* plants. Further research found *AtOZF2* participated in the ABA and salt stress responses through the *ABI2*-mediated signaling pathway [18]. Those pieces of evidence indicated that the GhC3H20 might interact with GhPP2CA and GhHAB1 to regulate salt stress by the ABA signaling pathway. As negative regulators participated in the ABA signaling pathway, *PP2CA* and *HAB1* were also involved in the regulation of stresses. PP2CA together with ABI1 and SnRK2.4 regulate root length under salt stress [40]. Located close to PP2CA, Type 2C Protein Phosphatases *SlPP2C1* gene RNAi plants displayed delayed root growth [41]. In this research, overexpression of *GhC3H20* enhanced root length in *Arabidopsis* seedlings under salt stress. Therefore, we speculated that GhC3H20 was involved in ABA signaling to regulate root growth by binding to GhPP2CA and GhHAB1 under salt stress.

## 4. Materials and Methods

### 4.1. Plant Material and Treatments

*G. hirsutum* standard line TM-1 was used in this study. Cotton seeds were planted in the sand and grown in a plant growth chamber at 25 °C with a 16 h light/8 h dark photoperiod condition. Until the third true leaf expanded for salt or drought treatment, seedlings were soaked in 200 mM sodium chloride (NaCl) or 20% polyethylene glycol 6000 (PEG6000) solution, respectively. Simultaneously, seedlings watered with the deionized water was used as a control. The three roots of the control and salt-treated seedlings were collected for qRT-PCR assay at each time point of 0, 1, 3, 6, 12, 24, and 48 h. The three leaves of the control and drought-treated seedlings were harvested for qRT-PCR assay at each time point of 0, 1, 3, 6, 12, 24, and 48 h. For ABA treatment, the 100 μM abscisic acid (ABA) solution was sprayed onto leaves. Samples from three leaves were collected at each time point of 0, 3, 6, 9, and 12 h. For the tissue-specific test, *G. hirsutum* standard line TM-1 was planted in the field. The root, stem, leaves, and buds were taken during the third leaf time. The petals, stamens, and pistils were taken during flowering. A total of 15 days of fiber were used to do this study. Then, samples were immediately frozen in liquid nitrogen and stored at −80 °C in a refrigerator for subsequent experiments.

### 4.2. DNA Extraction, RNA Isolation, and the qRT-PCR Analysis

The cetyl-trimethylammonium bromide (CTAB) method was used to extract genomic DNA as in the previous study [42]. The total RNA of leaves and roots was extracted and purified using an EASY Spin plus plant Total RNA Extraction kit (SunYa, Henan, China) or an RNAprep Pure Plant Kit (Tiangen, Beijing, China) according to the manufacturer’s instructions. First-strand synthesis of cDNA was performed by the PrimeScript™ RT Reagent kit with gDNA Eraser (Takara, Japan) or the HiScriptII Q RT SuperMix Vazyme for the qPCR (+gDNA wiper) (Vazyme, Nanjing, China) kit according to the manufacturer’s instructions. A cotton *actin* gene (*GhActin*) and an *Arabidopsis thaliana* polyubiquitin gene (*AtUBQ10*) were used as standard controls. All the primers used for qRT-PCR were designed using a primer database (http://biodb.swu.edu.cn/qprimerdb, last accessed on 26 February 2023) and are listed in Appendix A. The expression of genes was analyzed using a 7500 Real-Time PCR System (Applied Biosystems, Waltham, MA, USA) with the UltraSYBR Mixture (Low ROX) Kit. The 10 μL reaction volume contained the following components: 0.4 μL of the PCR forward primer (10 μM), 0.4 μL of the PCR reverse primer (10 μM), 1 μL of cDNA, 5 μL of SYBR Premix Ex Taq (2×), and 3.6 μL of ddH_2_O. The amplification parameters were as follows: 95 °C for 10 min, 40 cycles of 95 °C for 10 s, 60 °C for 30 s, and 72 °C for 32 s. The 2^−ΔΔCt^ method was applied to calculate the relative expression levels with three technical replicates [43].

### 4.3. Gene Clone and Sequence Analysis

The full-length coding sequences (CDS) and the 2000 bp upstream sequences of the *GhC3H20* (*GH_D08G2771*) gene from *G.hirsutum* TM-1 were cloned by specific primers (Appendix A). For the GUS assay, the *GhC3H20* promoter sequences were inserted into the PBI121 binary vector with HindIII and XbaI restriction sites to generate the Pro*GhC3H20*::GUS vector. For overexpression studies, the *GhC3H20* CDS was inserted into the pBI121 binary vector with *Xba*I and *Sac*I restriction sites to generate the 35S-*GhC3H20* vector. The vectors were transferred into the *Agrobacterium tumefaciens* strain GV3101 by the freezing and thawing method [44]. Multiple sequence alignments were performed using DNAMAN software, and a phylogenetic tree was conducted by MEGA 7.0 software using a neighbor-joining method. The *GhC3H20* gene structure was predicted by the online website Gene Structure Display Server (GSDS 2.0) (http://gsds.cbi.pku.edu.cn/). The molecular weight (Mw) and isoelectric point (pI) of the GhC3H20 protein were predicted by the online website ExPASy (http://web.expasy.org/compute_pi/). The online website ProtComp 9.0 (http://linux1.softberry.com/berry.phtml?topic=protcompan&group=programs&subgroup=proloc) was used to predict the subcellular localization of the GhC3H20 protein. The online website PlantCARE (http://bioinformatics.psb.ugent.be/webtools/plantcare/html/) was employed to predict the cis-elements.

### 4.4. Arabidopsis Transformation

The *Arabidopsis thaliana* Columbia ecotype was used for this study. The floral dip method was used to generate transgenic *Arabidopsis* plants [45]. Positive transformants were selected on 1/2 MS medium containing kanamycin (50 mg/l) and grew until maturation. Genomic DNA was extracted via the TPS method for PCR to test positive plants. The homozygous overexpressed transgenic lines of T_3_ generations were used for phenotypic analysis.

### 4.5. β-Glucuronidase (GUS) Histochemical Staining

The GUS staining kit (HuaYueYang, Bejing, China) was used for this experiment. For salt treatment, seeds of transgenic plants were placed on the 1/2 MS medium containing 150 mM NaCl. Samples from the roots, stems, leaves, inflorescences, and fruit pods of the T_1_ generation transgenic *Arabidopsis* were placed in the 5 mL tube to which the GUS staining solution was added. After 24 h at 37 °C in the dark, 75% alcohol was added to the tubes for decolorization.

### 4.6. Phenotypic Observation of Transgenic Arabidopsis

For length experiments, seeds of transgenic and WT *Arabidopsis* were germinated and grown on 1/2 MS medium supplemented with 0 (control), 150 mM NaCl, and 200 mM mannitol. After dark treatment at 4 °C for 2 days, the 1/2 MS medium plates with the seeds were placed in an incubator with a photoperiod of 16 h light/8 h dark at 22 °C. Ten seedlings were grown for two weeks. The root length in WT and transgenic lines were measured. Seedlings were transplanted in the nutrition soil and grown in a plant growth chamber at 25 °C with a 16 h light/8 h dark photoperiod condition. One-month-old plants were subjected to salt treatment. For salt treatment, the transgenic and WT plants were watered with 400 mM NaCl and then kept in the NaCl-contained soil for 5 days. The growth status of the transgenic and WT plants under high salinity (NaCl) was observed.

### 4.7. VIGS Assay

For the VIGS assay, the online website SGN VIGS Tool (https://vigs.solgenomics.net/?tdsourcetag=s_pcqq_aiomsg) was applied to obtain the silenced fragments of *GhC3H20*. The gene-specific primers were designed by the NCBI Primer-BLAST tool and are listed in Appendix A. The silenced fragments were cloned by PCR and integrated into the PYL156 binary vector to construct the PYL156-*GhC3H20* vector. The constructed vector and the PYL156 (negative control), PYL-*GhPDS* (positive control), and PLY192 (helper vector) vectors were then introduced into the *Agrobacterium tumefaciens* strain LBA4404. The LBA4404 cells harboring the vectors were collected and resuspended in filtration buffer (10 mM MgCl_2_, 10 mM MES, and 200 μM acetosyringone). The LBA4404 cells harboring PYL156 or PYL156-*GhC3H20* were equally mixed with PYL192 and co-injected into two fully expanded cotyledons of TM-1 plants. Until the third true leaves expanded, twenty seedlings were treated with 400 mM NaCl solution. The growth status of the PYL156 and PYL156 derivative plants under high salinity (NaCl) was observed.

### 4.8. Measurement of CAT and Chlorophyll Content

After salt treatment, approximately 0.1 g of samples from leaves were collected and immediately frozen in liquid nitrogen. The CAT content detection kit (Solarbio, Beijing, China) was used to measure the content of CAT. Samples were placed in the 15 mL tube to which 15 mL of mixed liquid (acetone: absolute ethanol = 1:1) was added and treated for 24 h at room temperature in the dark. When the leaves turned white, the absorbance was measured at 645 nm and 663 nm.
Chlorophyll Content (mg/g) = (20.31A_645_ + 8.03A_663_) × 0.15

### 4.9. Yeast Two-Hybrid (Y2H) Assays

The CDS sequences of *GhC3H20*, *GhPP2CA*, *GhHAB1*, *GhABF3*, and *GhABI1* were cloned and amplified by PCR. The fragments of CDS sequences were inserted into the pGBKT7 binary vector with *EcoR*I and *BamH*I sites to create pGBKT7-*GhC3H20*, pGBKT7-*GhPP2CA*, pGBKT7-*GhHAB1*, pGBKT7-*GhABF3*, and pGBKT7-*GhABI1* plasmids. The pGBKT7-*GhC3H20* with pGADT7, pGBKT7-*GhPP2CA*, pGBKT7-G*hHAB1*, pGBKT7-*GhABF3*, or pGBKT7-*GhABI1* plasmids were co-transformed into Y2HGold cells. The yeast cells containing transformed products, pGBKT7-p53+pGADT7-largeT (positive control), and pGBKT7-laminC+pGADT7-largeT (negative control) were cultured and detected on SD-Trp-Leu, SD-Trp-Leu-His-Ade containing 0, 20, 40, 60, 80, 100 mM 3AT, and SD-Trp-Leu-His-Ade+X-a-Gal+40 mM 3AT medium at 30 °C for 3–5 days.

## 5. Conclusions

As one of the transcription factors widely existing in plants, the CCCH zinc-finger protein not only plays pivotal roles in plant growth and development but also participates in stress response. In this study, a CCCH zinc-finger protein, the *GhC3H20* gene, was identified. qRT-PCR results showed that GhC3H20 was up-regulated under NaCl, PEG, and ABA treatments. GUS activity was detected in the root, stem, leaves, and flowers of Pro*GhC3H20*::GUS transgenic *Arabidopsis*. Compared with the control, the GUS activity of Pro*GhC3H20*::GUS transgenic *Arabidopsis* seedlings under NaCl treatment was stronger. Overexpressed *GhC3H20* enhanced the salt stress tolerance in *Arabidopsis*, and silencing of *GhC3H20* decreased the salt stress tolerance in cotton. In the Y2H assay, two interact genes (GhPP2CA and GhHAB1) of GhC3H20 were identified. *GhPP2CA* and *GhHAB1* are the key genes involved in the ABA signaling pathway. The expression levels of *PP2CA* and *HAB1* in transgenic *Arabidopsis* were higher than those in the WT, and pYL156-*GhC3H20* plants were lower than those in the control. GhC3H20 is likely to participate in the ABA signaling pathway to regulate salt stress by interacting with GhPP2CA and GhHAB1. This study lays the foundation for future studies of *GhC3H20* in the improvement of salt stress tolerance in cotton. For further study, transgenic cotton of *GhC3H20* will be constructed to investigate the biological function in response to salt stress. What is more, the downstream and upstream genes of *GhC3H20* will be further studied to form a regulatory network in response to salt stress.

## Figures and Tables

**Figure 1 ijms-24-05057-f001:**
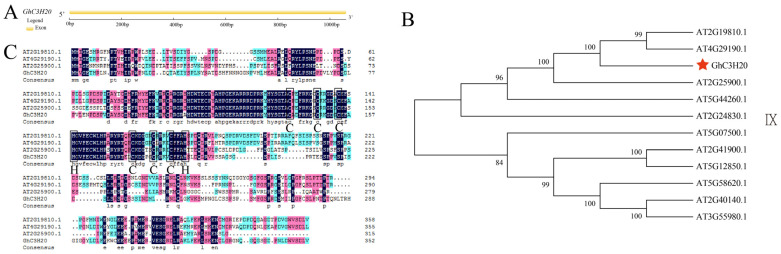
*GhC3H20* gene structure, sequence alignment, and phylogenetic analysis. (**A**) Gene structure of *GhC3H20*. (**B**) Phylogenetic analysis of *GhC3H20* and CCCH zinc-finger family group IX members in *Arabidopsis thaliana*. (**C**) Protein sequence alignments of GhC3H20 with AT2G19810, AT4G29190, and AT2G25900. Note: CCCH represents the C3H-type motif. The colored regions represent conserved amino acid sequences. The red star represents *GhC3H20* gene.

**Figure 2 ijms-24-05057-f002:**
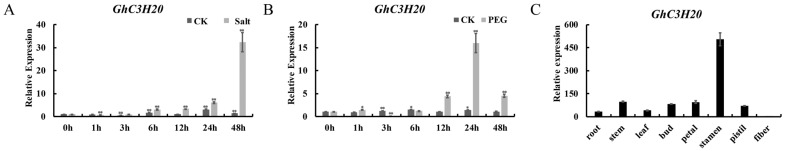
Expression pattern analysis of *GhC3H20* under 200 mM NaCl and 20% PEG treatment and in eight cotton tissues (root, stem, leaf, bud, petal, stamen, pistil, and fiber) in cotton. (**A**) Expression pattern analysis of *GhC3H20* gene in roots under ddH_2_O (CK) and 200 mM NaCl treatments. (**B**) Expression pattern analysis of *GhC3H20* gene in leaves under ddH_2_O (CK) and 20% PEG treatments. (**C**) Expression pattern analysis of *GhC3H20* gene in eight tissues. The error bars represent standard deviations of three technical replicates (* *p* < 0.05, ** *p* < 0.01, Student’s *t*-test).

**Figure 3 ijms-24-05057-f003:**
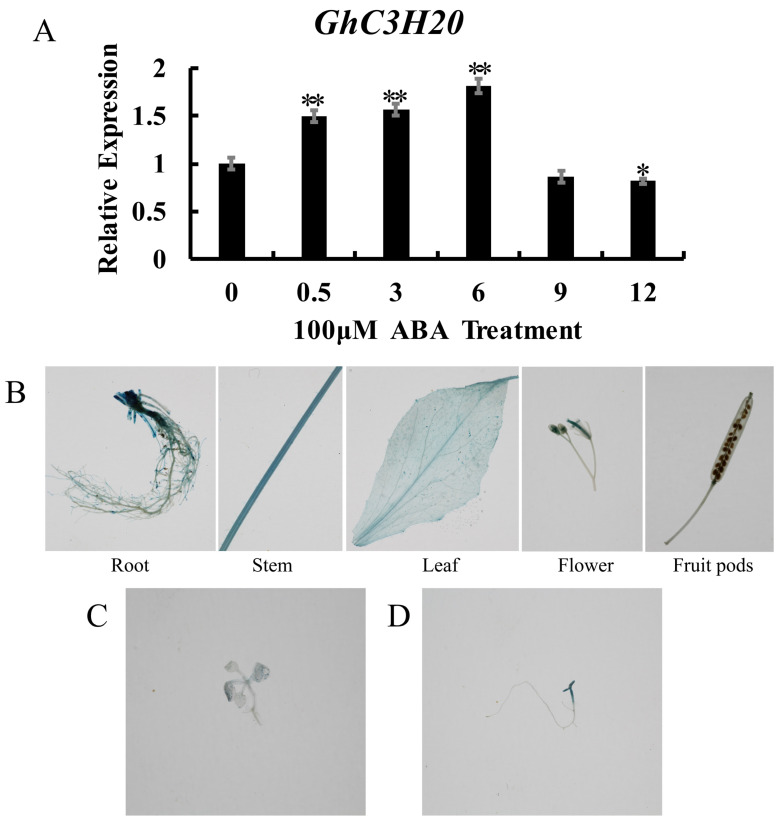
The expression level analysis of the *GhC3H20* gene under 100 μM ABA treatment. GUS activity analysis of Pro*GhC3H20*::GUS transgenic *Arabidopsis* in five tissues (root, stem, leaf, flower, and fruit pod), and GUS activity analysis of Pro *GhC3H20*::GUS transgenic *Arabidopsis* seedlings under control and salt treatment. (**A**) The expression level analysis of the *GhC3H20* gene under 100 μM ABA treatment. (**B**) GUS activity analysis of Pro*GhC3H20*::GUS transgenic *Arabidopsis* in five tissues (root, stem, leaf, flower, and fruit pod). (**C**) GUS activity analysis of Pro*GhC3H20*::GUS transgenic *Arabidopsis* seedlings on 1/2 MS medium. (**D**) GUS activity analysis of Pro*GhC3H20*::GUS transgenic *Arabidopsis* seedlings on 1/2 MS medium containing 150 mM NaCl. The error bars represent standard deviations of three technical replicates (* *p* < 0.05, ** *p* < 0.01, Student’s *t*-test).

**Figure 4 ijms-24-05057-f004:**
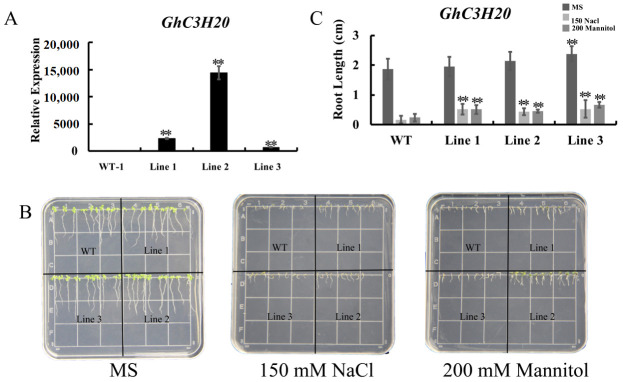
Overexpression of *GhC3H20* enhanced salt and osmotic stresses tolerance in *Arabidopsis* seedlings. (**A**) The expression levels of *GhC3H20* in WT *Arabidopsis* and three transgenic *Arabidopsis* lines. (**B**) Root length phenotype of WT *Arabidopsis* and three transgenic *Arabidopsis* lines under 1/2 MS medium, 150 mM NaCl, and 200 mM mannitol treatments. (**C**) Statistics of root lengths of *Arabidopsis thaliana* seedlings under salt and osmotic stresses. The error bars represent standard deviations of three technical replicates or standard deviations of the root length among *Arabidopsis* seedlings (** *p* < 0.01, Student’s *t*-test). Note: WT represents wild-type *Arabidopsis*. Line 1, Line 2, and Line 3 represents three transgenic *Arabidopsis* lines.

**Figure 5 ijms-24-05057-f005:**
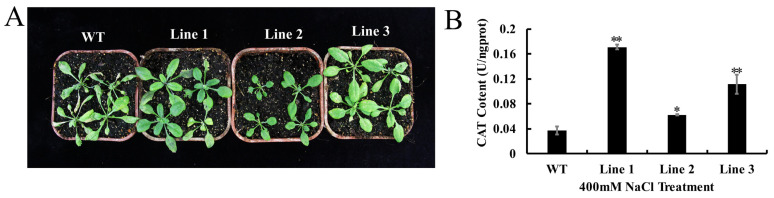
Phenotype and the content of CAT in WT *Arabidopsis* and three transgenic *Arabidopsis* lines under 400 mM NaCl treatment. (**A**) The phenotype of WT *Arabidopsis* and three transgenic *Arabidopsis* lines under 400 mM NaCl treatment. (**B**) The content of CAT in WT *Arabidopsis* and three transgenic *Arabidopsis* lines under 400 mM NaCl treatment. The error bars represent standard deviations of three biological replicates (* *p* < 0.05, ** *p* < 0.01, Student’s *t*-test).

**Figure 6 ijms-24-05057-f006:**
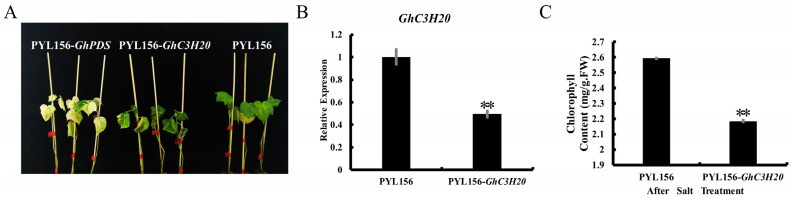
Silencing of *GhC3H20* decreased salt stress tolerance in cotton. (**A**) Leaf whitening of positive control plants (plants injected with pYL156-*GhPDS* vector), and phenotype analysis of control plants and silent plants under 400 mM NaCl treatment. (**B**) The relative expression levels of control plants and silenced plants in cotton leaves. (**C**) The content of chlorophyll in leaves of control plants and silenced plants under 400 mM NaCl treatment. The error bars represent standard deviations of three biological replicates (** *p* <0.01, Student’s *t*-test).

**Figure 7 ijms-24-05057-f007:**
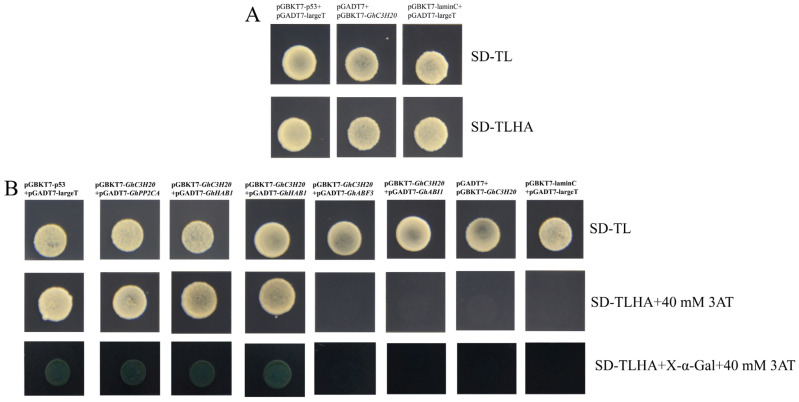
GhC3H20 interacted with GhPP2CA and GhHAB1 in vivo. (**A**) GhC3H20 transcriptional activation assay. (**B**) GhC3H20 interacted with GhPP2CA and GhHAB1 in yeast cells.

**Figure 8 ijms-24-05057-f008:**
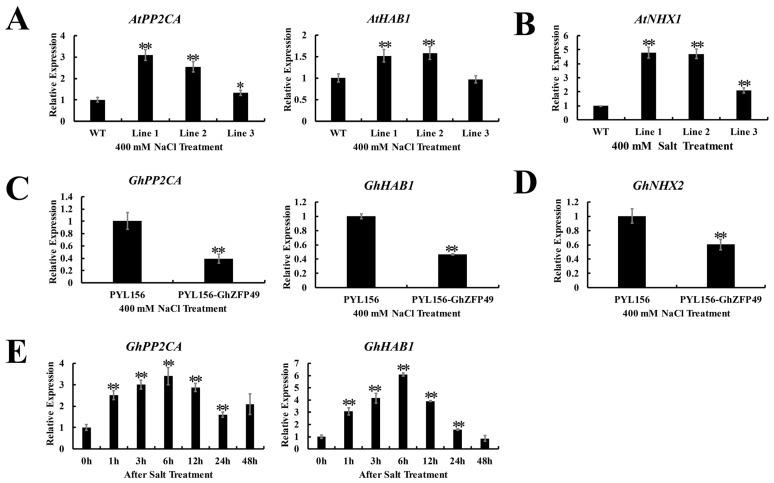
The expression levels of genes related to ABA and osmotic stress in cotton and *Arabidopsis* under 400 mM NaCl treatment. (**A**) The expression levels of ABA marker genes (*AtPP2CA* and *AtHAB1*) in WT *Arabidopsis* and three transgenic *Arabidopsis* lines in leaves under 400 mM NaCl treatment. (**B**) The relative expression of the osmotic stress marker gene (*AtNHX1*) in WT *Arabidopsis* and three transgenic *Arabidopsis* lines in leaves under 400 mM NaCl treatment. (**C**) The relative expression of ABA marker genes (*GhPP2CA* and *GhHAB1*) in control plants (plants injected with pYL156 empty vector) and silenced plants (plants injected with pYL156-*GhC3H20* vector) in leaves under 400 mM NaCl treatment. (**D**) The relative expression of the osmotic stress marker gene (*GhNHX2*) in control plants (plants injected with pYL156 empty vector) and silenced plants (plants injected with pYL156-*GhC3H20* vector) in leaves under 400 mM NaCl treatment. (**E**) The expression levels of *GhPP2CA* and *GhHAB1* in roots under 200 mM NaCl treatment. The error bars represent standard deviations of three technical replicates (* *p* < 0.05, ** *p* < 0.01, Student’s *t*-test).

## Data Availability

The data presented in this study are available on request from the corresponding authors.

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
