# Peer review of "The CCCH-Type Zinc-Finger Protein GhC3H20 Enhances Salt Stress Tolerance in Arabidopsis thaliana and Cotton through ABA Signal Transduction Pathway"

_ijms, 2023, doi:10.3390/ijms24055057_

Round 1
Reviewer 1 Report
The manuscript “The CCCH-type zinc-finger protein GhC3H20 enhances salt stress tolerance in Arabidopsis thaliana and cotton through ABA signal transduction pathway" by Zhang et al. thoroughly characterized a potential gene GhC3H20 in cotton and Arabidopsis for salt tolerance. Authors have verified the roles of GhC3H20 through various potential analyses such as expression pattern, GUS activity and genetic transformation. Overexpression of GhC3H20 in Arabidopsis increases salt stress tolerance while target gene silencing in cotton compromised salt stress tolerance. Further interacting proteins were identified are verified related to ABA signaling pathway. The results are well presented and discussion is logically sound. Overall this work provides valuable information and concepts about roles of GhC3H20 gene in cotton. With these comments, I recommend acceptance.
Author Response
Dear reviewer:
On behalf of my co-authors, we thank you very much for your helpful efforts processing our manuscript entitled "The CCCH-type zinc-finger protein GhC3H20 enhances salt stress tolerance in Arabidopsis thaliana and cotton through ABA signal transduction pathway" (Manuscript ID: ijms-2190845). With regard to your positive and constructive comments on our manuscript, we think they are of high value and importance as well as of critical guiding significance to our researches.

Reviewer 2 Report
The manuscript describes the function of GhC3H20 from cotton in salt tolerance and its relationship to abscisic acid signaling. The manuscript is well written, and the conclusion is well supported by the gain-of-function and loss-of-function analyses in Arabidopsis and cotton. Thus, the manuscript is acceptable for publication after addressing only one suggestion.
Please consider describing the homology PP2CA and HAB1 between Arabidopsis and cotton. In the transgenic Arabidopsis, AtPP2CA and AtHAB1 are analyzed. Are these genes orthologous to GhPP2CA and GhHAB1?
Author Response
Dear reviewer:
On behalf of my co-authors, we thank you very much for your helpful efforts processing our manuscript entitled "The CCCH-type zinc-finger protein GhC3H20 enhances salt stress tolerance in Arabidopsis thaliana and cotton through ABA signal transduction pathway" (Manuscript ID: ijms-2190845) and providing us an opportunity to revise it. With regard to your positive and constructive comments on our manuscript, we think they are of high value and importance for revising and improving our paper, as well as of critical guiding significance to our researches.
After careful reviewing on your comments, we have made correspondent revisions in the manuscript, which we hope will meet the requirements of your journal. Both of the tractable and clean versions of the revised manuscript were submitted for your convenience.
Thank you again for you kind help and efforts. If there is anything that need us do, please do not hesitate to let us know.
And here are my sincere responses.
Please consider describing the homology PP2CA and HAB1 between Arabidopsis and cotton. In the transgenic Arabidopsis, AtPP2CA and AtHAB1 are analyzed. Are these genes orthologous to GhPP2CA and GhHAB1?
Thank you very much for your suggestion. In our revised manuscript, we have introduced them in the article.
Lines 245-246, Page 8.

Reviewer 3 Report
In this Manuscript, based on the transcriptome data related to salt stress, an up-regulated CCCH zinc finger gene GhC3H20 was identified in this experiment. qRT-PCR results showed that GhC3H20 was up-regulated under NaCl, PEG, and ABA treatments. Compared with the control, the GUS activity of ProGhC3H20::GUS transgenic Arabidopsis seedlings under NaCl treatment was stronger. Overexpression of GhC3H20 in Arabidopsis could promote salt stress tolerance and silencing of GhC3H20 in cotton can decrease salt stress tolerance. In addition, GhC3H20 could interact with GhPP2CA and GhHAB1 to regulate salt stress by the ABA signaling pathway. This study lays a foundation for future studies of the GhC3H20 in the improvement of salt stress tolerance in cotton.
It is a very complete job; I enjoyed reading it. I only have two comments:
-Many sentences in the introduction part have missed the appropriate citations (e.g., Lines 37, 40, 41, 45, 72, 79, etc.). Please provide appropriate references.
-Conclusion was not well written. It is too short. This part should be improved. Future studies should be discussed.
Author Response
Dear reviewer:
On behalf of my co-authors, we thank you very much for your helpful efforts processing our manuscript entitled "The CCCH-type zinc-finger protein GhC3H20 enhances salt stress tolerance in Arabidopsis thaliana and cotton through ABA signal transduction pathway" (Manuscript ID: ijms-2190845) and providing us an opportunity to revise it. With regard to your positive and constructive comments on our manuscript, we think they are of high value and importance for revising and improving our paper, as well as of critical guiding significance to our researches.
After careful reviewing on your comments, we have made correspondent revisions in the manuscript, which we hope will meet the requirements of your journal. Both of the tractable and clean versions of the revised manuscript were submitted for your convenience.
Thank you again for you kind help and efforts. If there is anything that need us do, please do not hesitate to let us know.
And here are my sincere responses.
- Many sentences in the introduction part have missed the appropriate citations (e.g., Lines 37, 40, 41, 45, 72, 79, etc.). Please provide appropriate references.
Thank you very much for your suggestion. In our revised manuscript, we have added the appropriate citations in the article.
Lines 39-48 and 81, Page 1-2.
- Conclusion was not well written. It is too short. This part should be improved. Future studies should be discussed.
Thank you very much for your suggestion. In our revised manuscript, we have revised the discussion in the article.
Lines 448-466, Page 13.

Reviewer 4 Report
This study examined the role of a specific cotton gene, GhC3H20, in response to salinity or osmotic stress in cotton and transgenic Arabidopsis OX and VIGs lines.
My comments on this paper are offered below:
Line 21: Write out CAT (catalase) the first time you use it
LIne 38-89- Need a reference citing importance of cotton and losses due to salt.
Line 41- This sentence is vague and needs to be more speciifc (which TFs)
Line 49-51- This is a sentence fragment
Lines 57-60 These references seem irrelevant to this study. Please use more relevant references
Line 81- Reference needed. Is this in reference to this article or one where you used transcriptome data to find ChC3H20..if so, cite that paper.
Line 86-reword to past tense if you are describing this study (instead of using "could')
Line 98- Nucleus, not nuclear
Figure 2: You are missing the control treatments (no salt or PEG) to show the constitutive levels of GhC3H20 without treatments. Only by doing this can you say that the gene is salt or PEG induced. Also, for 2C, at what time point (how long after treatment) did you test gene expression in tissues?
***All figures***: What do your error bars represent and how many replicates, as well as PCR technical replicates did you have for each experiment? This is not indicated in the methods or anywhere in the captions of the figures and should be made very clear.
Lines 140-145 are redundant.
Figure 3B- A B C font are very large. Please label FIgure 3B as 3Ba or 3B1, 3B2 or indicate right to left labels in the captions for each tissue type.
Figure 4D- There are. numerous methods and models (published) that have been used for measuring root length. How did you do this? This is not in the methods? How were (if any) lateral roots included?
Figure 5- How did you measure Rosette size or chlorosis? THis is not clear in the methods By observation alone is not sufficient. Line 2 looks smaller than WT.
Figure 6: Why is there no water or untreated control in the figures? Did they grow that way without NaCl? Just having a "control/no VIGS" plant does not give you a control for this study.
Lines 259-260 This statement is repeated many times throughout the article but is not elaborated upon. Needs additional citations or details.
Conclusions- this section is weak and just vaguely repeats what has already been said in the discussion. It is unnecessary.
Author Response
Dear reviewer:
On behalf of my co-authors, we thank you very much for your helpful efforts processing our manuscript entitled "The CCCH-type zinc-finger protein GhC3H20 enhances salt stress tolerance in Arabidopsis thaliana and cotton through ABA signal transduction pathway" (Manuscript ID: ijms-2190845) and providing us an opportunity to revise it. With regard to your positive and constructive comments on our manuscript, we think they are of high value and importance for revising and improving our paper, as well as of critical guiding significance to our researches.
After careful reviewing on your comments, we have made correspondent revisions in the manuscript, which we hope will meet the requirements of the journal.
And here are my sincere responses.
- Line 21: Write out CAT (catalase) the first time you use it.
Response:
Thank you very much for your careful and valuable reviewing. In our revised manuscript, we have added it in the article.
Lines 24, Page 1.
- Line 38-89- Need a reference citing importance of cotton and losses due to salt.
Response:
In our revised manuscript, we have added the appropriate citations in the article.
Lines 41-42, Page 1.
- Line 41- This sentence is vague and needs to be more speciifc (which TFs).
Response:
In our revised manuscript, we have added it in the article.
Lines 44-45, Page 2.
- Line 49-51- This is a sentence fragment.
Response:
In our revised manuscript, we have revised this sentence in the article.
Lines 51-54, Page 2.
- Lines 57-60 These references seem irrelevant to this study. Please use more relevant references
Response:
In our revised manuscript, we have deleted this part in the article.
- Line 81- Reference needed. Is this in reference to this article or one where you used transcriptome data to find ChC3H20..if so, cite that paper.
Response:
In our revised manuscript, we have added the appropriate citation in the article.
Lines 77, Page 2.
- Line 86-reword to past tense if you are describing this study (instead of using "could')
Response:
In our revised manuscript, we have revised “can” to “could” in the article.
Lines 82, Page 2.
- Line 98- Nucleus, not nuclear
Response:
In our revised manuscript, we have revised “nuclear” to “nucleus” in the article.
Lines 93, Page 2.
- Figure 2: You are missing the control treatments (no salt or PEG) to show the constitutive levels of GhC3H20 without treatments. Only by doing this can you say that the gene is salt or PEG induced. Also, for 2C, at what time point (how long after treatment) did you test gene expression in tissues?
Response:
Thank you for your careful work. 0h treatment is the control treatment (no salt or PEG). Moreover, we also treated the cotton seedlings with ddH2O as Ck to compare the expression levels of GhC3H20 with salt or PEG treatment. For figure 2C, root, stem, leaf and bud were taken during the third leaf time. petal, stamen, pistil were taken during the flowering time. 15 days of fiber were used to do this study.
- ***All figures***: What do your error bars represent and how many replicates, as well as PCR technical replicates did you have for each experiment? This is not indicated in the methods or anywhere in the captions of the figures and should be made very clear.
Response:
In our revised manuscript, we have clarified these information in the article.
Lines 123-124, Page 4; Lines 151-152, Page 5; Lines 170-171, Page 6; Lines 185-186 and Lines 204-205, Page 7; Lines 257-258, Page 9.
- Lines 140-145 are redundant.
Response:
In our revised manuscript, we have revised this part in the article.
Lines 168-169, Page 6.
- Figure 3B- A B C font are very large. Please label FIgure 3B as 3Ba or 3B1, 3B2 or indicate right to left labels in the captions for each tissue type.
Response:
In our revised manuscript, we have revised the figure 3 in the article.
- Figure 4D- There are. numerous methods and models (published) that have been used for measuring root length. How did you do this? This is not in the methods? How were (if any) lateral roots included?
Response:
In this study, we measured the taproot length of Arabidopsis under salt and mannitol treatments according to the reference below.
Belachew, K.Y.; Stoddard, F.L. Screening of faba bean (Vicia faba L.) accessions to acidity and aluminium stresses. PeerJ 2017, 5, e2963, doi:10.7717/peerj.2963.
- Figure 5- How did you measure Rosette size or chlorosis? THis is not clear in the methods By observation alone is not sufficient. Line 2 looks smaller than WT.
Response:
In this study, we didn’t measure Rosette size or chlorosis.
For the “Line 2 looks smaller than WT.”, we put more attention on the degree of wither. The methods are as followed:
Seeds of WT and three lines of Arabidopsis were planted in the nutrient soil. When Arabidopsis unfolds its first true leaf, four plants are transferred to a new nutrient soil and continue to grow with the same condition. One-month-old plants were subjected to salt treatment. For salt treatment, the transgenic and WT plants were watered with 400 mM NaCl and then kept in the NaCl-contained soil for 5 days. Compare with WT, it seems that the leaves of three lines Arabidopsis are not so withered. What’s more, the content of CAT in WT was less than in three lines. In summary, overexpression of GhC3H20 enhanced the salt stress tolerance in Arabidopsis.
- Figure 6: Why is there no water or untreated control in the figures? Did they grow that way without NaCl? Just having a "control/no VIGS" plant does not give you a control for this study.
Response:
Add two group plants, including pYL156 and pYL156-GhC3H20 plants under water treatment, is obviously a better way to confirm the function of GhC3H20. However, when we design the experiment, we referenced many previous studies (as follows), which did not add these group that under the water treatment. Therefore, we believe our VIGS assays is a common practices.
Rong W, Qi L, Wang A, Ye X, Du L, Liang H, Xin Z, Zhang Z. The ERF transcription factor TaERF3 promotes tolerance to salt and drought stresses in wheat. Plant Biotechnol J. 2014 May;12(4):468-79. doi: 10.1111/pbi.12153. Epub 2014 Jan 3. PMID: 24393105.
Long L, Yang WW, Liao P, Guo YW, Kumar A, Gao W. Transcriptome analysis reveals differentially expressed ERF transcription factors associated with salt response in cotton. Plant Sci. 2019 Apr;281:72-81. doi: 10.1016/j.plantsci.2019.01.012. Epub 2019 Jan 18. PMID: 30824063.
Chen P, Wei F, Jian H, Hu T, Wang B, Lv X, Wang H, Fu X, Yu S, Wei H, Ma L. A Comprehensive Gene Co-Expression Network Analysis Reveals a Role of GhWRKY46 in Responding to Drought and Salt Stresses. Int J Mol Sci. 2022 Oct 12;23(20):12181. doi: 10.3390/ijms232012181. PMID: 36293038; PMCID: PMC9603583.
Wei F, Chen P, Jian H, Sun L, Lv X, Wei H, Wang H, Hu T, Ma L, Fu X, Lu J, Li S, Yu S. A Comprehensive Identification and Function Analysis of Serine/Arginine-Rich (SR) Proteins in Cotton (Gossypium spp.). Int J Mol Sci. 2022 Apr 20;23(9):4566. doi: 10.3390/ijms23094566. PMID: 35562957; PMCID: PMC9105085.
- Lines 259-260 This statement is repeated many times throughout the article but is not elaborated upon. Needs additional citations or details.
Response:
As the version of the manuscript in the system, the line 259-260 might refered to the part of “To elucidate the precise molecular mechanism of GhC3H20 increasing salt tolerance in transgenic plants and cotton, two proteins – GhPP2CA and GhHAB1 – that interact with GhC3H20 were isolated and identified by Y2H assay.”
We cited many papers responding to the GhPP2CA and GhHAB1 in line 270-272. We hope these citations could meet the demand.
- Conclusions- this section is weak and just vaguely repeats what has already been said in the discussion. It is unnecessary.
Response:
In our revised manuscript, we have revised the conclusion in the article.

Round 2
Reviewer 4 Report
The authors address most of my concerns, but there are still 3 major flaws which have not been addressed or corrected. The authors discuss significant differences among treatments, but still have not described their statistical methods and total sample sizes. In the abstract, the authors claim "Virus induced gene si-lencing (VIGS) experiment showed that compared with the control, the leaves of pYL156-GhC3H20 plants were wilted and dehydrated." However, there is still no method described on how they measured this. Observation alone is insufficient. Finally, there is no water control in their VIGS treatment. Even though they cite other papers that have done similar work, this study lacks a control.
Author Response
Dear reviewer:
On behalf of my co-authors, we thank you very much for your helpful efforts processing our manuscript entitled "The CCCH-type zinc-finger protein GhC3H20 enhances salt stress tolerance in Arabidopsis thaliana and cotton through ABA signal transduction pathway" (Manuscript ID: ijms-2190845) and providing us an opportunity to revise it. With regard to your positive and constructive comments on our manuscript, we think they are of high value and importance for revising and improving our paper, as well as of critical guiding significance to our researches.
After careful reviewing on your comments, we have made correspondent revisions in the manuscript, which we hope will meet the requirements of the journal.
And here are my sincere responses.
- The authors discuss significant differences among treatments, but still have not described their statistical methods and total sample sizes.
Response:
Thank you very much for your careful and valuable review. In our revised manuscript, we have added it in the article.
Lines 123-124, Page 4
The error bars represent standard deviations of three technical replicates (*P<0.05 ** P<0.01 Student’s t-test).
Lines 151-152, Page 5
The error bars represent standard deviations of three technical replicates (*P<0.05 ** P<0.01 Student’s t-test).
Lines 161 and 170-171, Page 6
Lines 161: the root length of ten Arabidopsis seedlings were
Lines 170-171: The error bars represent standard deviations of three technical replicates or standard deviations of the root length among Arabidopsis seedlings (** P<0.01 Student’s t-test).
Lines 185-186 and 204-205, Page 7
Lines 185-186 : The error bars represent standard deviations of three biological replicates(*P<0.05 ** P<0.01 Student’s t-test).
Lines 204-205: The error bars represent standard deviations of three biological replicates (** P<0.01 Student’s t-test).
Lines 344, 348, and 349-352, Page 11
Lines 344: The three roots of control and salt-treated seedlings were
Lines 348: Samples from three leaves were collected at each time
Lines 349-352: For tissue specific test, G. hirsutum standard line TM-1 was planted in the field. Root, stem, leaf and bud were taken during the third leaf time. petal, stamen, pistil were taken during the flowering time. 15 days of fiber were used to do this study.
Lines 409, Page 12
Ten seedlings were grown for two weeks.
Lines 428, Page 13
expanded, twenty seedlings were treated with 400 mM NaCl solution.
- In the abstract, the authors claim "Virus induced gene si-lencing (VIGS) experiment showed that compared with the control, the leaves of pYL156-GhC3H20 plants were wilted and dehydrated." However, there is still no method described on how they measured this. Observation alone is insufficient.
Response:
Thank you very much for your careful and rigorous review. By observing the phenotype of the silenced plants and the control plants, we found that the leaves of the silenced plants wilted and turned yellow, while the leaves of the control plants only slightly shrank due to water loss. In addition to the observation of biological phenotype, some gene function studies (as follows) also measured the content of chlorophyll to reflect the stress tolerance of plants.
Ma L, Hu L, Fan J, Amombo E, Khaldun ABM, Zheng Y, Chen L. Cotton GhERF38 gene is involved in plant response to salt/drought and ABA. Ecotoxicology. 2017 Aug;26(6):841-854. doi: 10.1007/s10646-017-1815-2. Epub 2017 May 23. PMID: 28536792.
Li Y, Feng Z, Wei H, Cheng S, Hao P, Yu S, Wang H. Silencing of GhKEA4 and GhKEA12 Revealed Their Potential Functions Under Salt and Potassium Stresses in Upland Cotton. Front Plant Sci. 2021 Dec 7;12:789775. doi: 10.3389/fpls.2021.789775. PMID: 34950173; PMCID: PMC8689187.
Rong W, Qi L, Wang A, Ye X, Du L, Liang H, Xin Z, Zhang Z. The ERF transcription factor TaERF3 promotes tolerance to salt and drought stresses in wheat. Plant Biotechnol J. 2014 May;12(4):468-79. doi: 10.1111/pbi.12153. Epub 2014 Jan 3. PMID: 24393105.
To further verify our observations, the leaves of the silenced plants and the control plants were taken after NaCl treatment. The content of chlorophyll was measured between the silenced plants and the control plants. The content of chlorophyll decreased after leaf wilting and water loss. Our observation was further confirmed by the determination of chlorophyll content. We know that it is not enough to only measure the content of chlorophyll to reflect the salt stress tolerance of plants. Some physiological indicators such as leaf water loss rate should be added to better reflect the results of this experiment. Through your review, we deeply realize that we still have many deficiencies in the experimental design. We sincerely thank you for the questions you raised in this draft review. We will correct these problems in future experiments to make our experimental design more rigorous. Finally, thank you again for your valuable comments on this review.
- Finally, there is no water control in their VIGS treatment. Even though they cite other papers that have done similar work, this study lacks a control.
Response:
Thank you very much for your careful and rigorous review. VIGS technology is a better way to study the function of GhC3H20 under salt stress. In this study, we used the VIGS to silence the GhC3H20 gene and study the biological function under salt stress. According to many previous studies (as follows), we design the experiment.
Rong W, Qi L, Wang A, Ye X, Du L, Liang H, Xin Z, Zhang Z. The ERF transcription factor TaERF3 promotes tolerance to salt and drought stresses in wheat. Plant Biotechnol J. 2014 May;12(4):468-79. doi: 10.1111/pbi.12153. Epub 2014 Jan 3. PMID: 24393105.
Long L, Yang WW, Liao P, Guo YW, Kumar A, Gao W. Transcriptome analysis reveals differentially expressed ERF transcription factors associated with salt response in cotton. Plant Sci. 2019 Apr;281:72-81. doi: 10.1016/j.plantsci.2019.01.012. Epub 2019 Jan 18. PMID: 30824063.
Chen P, Wei F, Jian H, Hu T, Wang B, Lv X, Wang H, Fu X, Yu S, Wei H, Ma L. A Comprehensive Gene Co-Expression Network Analysis Reveals a Role of GhWRKY46 in Responding to Drought and Salt Stresses. Int J Mol Sci. 2022 Oct 12;23(20):12181. doi: 10.3390/ijms232012181. PMID: 36293038; PMCID: PMC9603583.
Wei F, Chen P, Jian H, Sun L, Lv X, Wei H, Wang H, Hu T, Ma L, Fu X, Lu J, Li S, Yu S. A Comprehensive Identification and Function Analysis of Serine/Arginine-Rich (SR) Proteins in Cotton (Gossypium spp.). Int J Mol Sci. 2022 Apr 20;23(9):4566. doi: 10.3390/ijms23094566. PMID: 35562957; PMCID: PMC9105085.
These studies used the empty vector as the control under salt stress. The phenotype of silenced plants and control plants were observed after salt treatment. Compared with the control, the phenotype of silenced plants were sensitive to salt stress. Although the research we refer to and our research have empty carrier plants as a control, our experiment needs to be improved according to the questions you raised in the draft review. If we have water control or plants before treatment as control, our research will be more rigorous and the results will be more reliable. In the follow-up research, we should improve our experimental methods and make our experiments more rigorous. Finally, thank you again for your valuable comments on this review.

Round 3
Reviewer 4 Report
The authors have addressed my concerns with regards to the experimental design and number of replicates and statistics.